# Himawari-8/AHI Aerosol Optical Depth Detection Based on Machine Learning Algorithm

Yuanlin Chen [1,2,3] , Meng Fan [2,*], Mingyang Li [1,2], Zhongbin Li [2], Jinhua Tao [2], Zhibao Wang [3] and Liangfu Chen [1,2]

1   University of Chinese Academy of Sciences, Beijing 100101, China; chenyuanlin@aircas.ac.cn (Y.C.); limyoung@aircas.ac.cn (M.L.); chenlf@aircas.ac.cn (L.C.)
2   State Key Laboratory of Remote Sensing, Aerospace Information Research Institute, Chinese Academy of Sciences (AIR-CAS), Beijing 100101, China; lizhongbin@aircas.ac.cn (Z.L.); taojh@aircas.ac.cn (J.T.)
3   School of Computer & Information Technology, Northeast Petroleum University, Daqing 163318, China; wangzhibao@nepu.edu.cn
*   Correspondence: fanmeng@aircas.ac.cn

**Abstract:** Due to the advantage of geostationary satellites, Himawari-8/AHI can provide near-real-time air quality monitoring over China with a high temporal resolution. Satellite-based aerosol optical depth (AOD) retrieval over land is a challenge because of the large surface contribution to the top of atmosphere (TOA) signal and the uncertainty of aerosol modes. Here, by combining satellite TOA reflectance, sun-sensor geometries, meteorological factors and vegetation information, we propose a data-driven AOD detection algorithm based on a deep neural network (DNN) model for Himawari-8/AHI. It is trained by sample data of 2018 and 2019 and is applied to derive hourly AODs over China in 2020. By comparison with ground-based AERONET measurements, $R^2$ for DNN-estimated AOD is up to 0.8702, which is much higher than that for the AHI AOD product with $R^2 = 0.4869$. The hourly AOD results indicate that the DNN model has a good potential in improving the performance of AOD retrieval in the early morning and in the late afternoon, and the spatial distribution is reliable for capturing the variation of aerosol pollution on the regional scale. By analyzing different DNN modeling strategies, it is found that seasonal modeling can hardly increase the accuracy of AOD retrieval to a certain extent, and $R^2$ increases from 0.7394 to 0.8168 when meteorological features, especially air pressure, are involved in the model training.

**Keywords:** AOD; DNN; AHI; Himawari-8; machine learning



## 1. Introduction

Atmospheric aerosols are solid and liquid particles suspended in the atmosphere, which are mainly emitted from natural and anthropogenic sources [1]. They play a key role in the Earth's radiation balance by directly scattering and absorbing solar radiation and indirectly changing the microphysical and radioactive properties of clouds [2]. Aerosol particulate matter with a diameter less than 2.5 μm ($PM_{2.5}$) is the main pollutant threatening health [3,4]. It was found that there is a strong positive relationship between ground-level $PM_{2.5}$ concentration and satellite-measured AOD [5–7]. Therefore, promoting the accuracy of satellite-retrieved AOD can make the performance of the $PM_{2.5}$ estimation better.

Although accurate and continuous aerosol information can be available through ground observation, AOD measured by sparsely distributed sites cannot fill the gap without satellite observation. Space-based sensors that can be used to retrieve aerosols include the Moderate Resolution Imaging Spectroradiometer (MODIS) onboard the TERRA and AQUA satellites, the Visible Infrared Imaging Radiometer Suite (VIIRS) onboard the Suomi-NPP and NOAA20 satellites [8], the Medium Resolution Spectral Imager (MERSI) onboard the FengYun Series Satellites [9], the Multi-angle Imaging SpectroRadiometer (MISR) [10],

the Polarization and Directionality of the Earth's Reflectance (POLDER) [11], the Advanced Very High Resolution Radiometer (AVHRR) [12], the Ozone Monitoring Instrument (OMI) [13] and the Sea-viewing Wide Field-of-view Sensor (SeaWiFS) [14,15]. All these sensors are carried on the sun-synchronous satellites, and their overpass time is relatively concentrated (10:30~13:30 Local Time). However, the vertical and horizontal distributions of aerosol in the atmosphere are related to the diurnal variation of the boundary layer height, meteorological and other factors. The loading and spatiotemporal distribution of aerosol change significantly in one day. Therefore, the temporal resolution does not allow capturing the diurnal variation of aerosols, which is an important feature of atmospheric environment monitoring and climate research.

In recent years, many geostationary satellite sensors were launched to monitor the global aerosol loading. The radiometric, spectral and spatial resolution of new-generation geostationary satellites are higher than those previously available in geostationary orbit. Himawari-8, a next-generation geostationary meteorological satellite successfully launched in 2014, carries a new multi-wavelength imager named Advanced Himawari imager (AHI). AHI can provide near-real-time observations over the East Asia and Western Pacific regions with a high temporal resolution of 10 min for a full disk. The Advanced Baseline Imager (ABI) carried on the Geostationary Operational Environmental Satellite (GOES) scans the full disk every 10 min in the default mode, with the coverage of most areas of America, the Pacific and the Atlantic [16]. The Spinning Enhanced Visible and Infrared Imager (SEVIRI) onboard the Meteosat Second Generation (MSG) scans the full disk every 15 min with the coverage of most areas of Europe, Africa and the Indian Ocean [17]. The Advanced Geosynchronous Radiation Imager (AGRI) is a primary optical instrument onboard Fengyun-4 series of satellites (the second-generation geostationary meteorological satellites in China). Its full-disk scanning time and regional scanning time are 15 and 1 min, respectively [18]. Moreover, the Earth Polychromatic Imaging Camera (EPIC) onboard the Deep Space Climate Observatory (DSCOVR) scans the entire sunlight side of the Earth every 1~2 h [19]. This study focuses on the Himawari-8/AHI, which has been frequently used for the dynamic monitoring of regional aerosol loading and particulate pollutants.

Satellite-based AOD retrieval over land is a challenge because of the large surface contribution to the top of atmosphere (TOA) signal for "bright" targets (i.e., urban, desert and snow/ice-covered surfaces with high surface reflectance) [20] and the uncertainty of selected aerosol type led by the temporal and spatial variability of aerosol concentration and the complex mixing of aerosol types [21–23]. Therefore, it is necessary to effectively separate the contribution of surface reflectance and atmospheric path radiation from the TOA measurement. In addition, satellite observations cannot provide sufficient information on aerosol properties, so the aerosol type must be assumed in the AOD retrieval procedure.

The accuracy of the aerosol contribution separated from TOA signals largely depends on the accurate measurement of surface reflection [24,25]. Currently, the Dark Target (DT) algorithm [26,27] and the Deep Blue (DB) algorithm [14,28,29] are the most widely used satellite AOD retrieval methods. The band ratio relationship of the DT method is not suitable for bright surface areas with high surface reflectance such as urban and bare land, and the relationship between the short-wave infrared (SWIR) surface reflectance of red and blue channels is not fixed due to the different spectral reflectance for different periods and surface targets. The DB algorithm can be used to retrieve the AOD of desert, urban and bare land, but its inversion accuracy is lower than that of the DT method [30]. Ge et al. retrieved Himawari-8/AHI AOD by using the relationship of surface reflectance between the visible and SWIR bands and following the DT algorithm [31]. However, considering the observation duration of a geostationary satellite, there is a limitation for this method to establish the relationship between the surface reflectance in the visible spectrum band and the TOA reflectance in the SWIR band. Yu et al. compared Himawari-8/AHI and Suomi-NPP/VIIRS data and found that they had a high consistency in the bands used for AOD inversion, indicating that the spatial resolution and the calibration accuracy of the Himawari-8/AHI were very high [32]. Yoshida et al. retrieved the Himawari-8/AHI AOD

based on the minimum apparent reflection and the DT algorithm, and their inversion results (R = 0.56, RMSE = 0.174) showed that the algorithm still needs to be improved by verifying based on the observation data of two AErosol RObotic NETwork (AERONET) stations in South Korea (Baeksa and Tahwa sites) [33]. Zhang et al. evaluated the performance of the Japan Aerospace Exploration Agency (JAXA) Himawari-8/AHI AOD product over China, indicating that the deviation of the AOD product largely depends on the season and surface type [34]. According to the study of Gao et al., in the aspect of temporal variation, an overestimation from JAXA Himawari-8/AHI AOD occurred in the afternoon for bare surface while an underestimation occurred in the morning for well-vegetation surfaces over China [35].

Some unknown error sources, such as nonlinear distribution of the dataset and abnormal distribution of data, can also introduce uncertainties into AOD retrievals. Therefore, the commonly used linear method is not sufficient to estimate the aerosol characteristics. An effective way to eliminate unnecessary situations is to use nonparametric methods, e.g., machine learning (ML) techniques. ML is a branch of artificial intelligence. It generates algorithms that can learn from the behavior or characteristics of specific datasets. Through these methods, the learning dataset is divided into "input" and "output", and an algorithm is developed to find the relationship between them. The outputs are the variables that must be predicted, and the inputs are the variables that provide information for the algorithm [36]. ML algorithms perform well in deriving this correlation, especially on technical issues [37–39], and they can discover the potential behavior of a system from long-term observation [40]. In recent years, the ML methods were gradually used for air quality prediction (e.g., AOD and ground-level $PM_{2.5}$, $NO_2$ and $O_3$ concentrations) [41–46]. Among various kinds of ML methods, the Artificial Neural Network (ANN) as a widely used machine learning method is famous for optimization. Some previous studies showed that ANN method had a good potential in improving the satellite-retrieved AOD modeling and AOD bias correction. Kollos et al. estimated AOD during dust outbreaks by using an ANN model and SEVIRI data, with a high correlation coefficient of 0.91 and a low mean absolute error (MAE) of 0.031 [47]. Lanzaco et al. applied both ANN and support vector machines to obtain corrected MODIS AOD values. Fan et al. developed an algorithm based on the ANN method driven by coupled atmosphere–ocean radiative transformation simulations to derive the MERSI-II/Fengyun-3D AODs, with an average percentage error around 23–29% for all bands [48].

In this study, the Deep Neural Network (DNN) will be used to improve AOD retrieval instead of classically physical algorithms (e.g., BT and DB algorithms). The DNN model is an ANN with multiple layers between the input and output layers and has a huge and complex modeling ability. Here, we tried to directly establish a nonlinear relationship between AOD and multi-factors (i.e., TOA reflectance, sun-sensor geometries, meteorological factors and vegetation information) to improve the adaptability of the AOD inversion model under complex conditions. Section 2 describes the satellite observations, ground-based measurements and model reanalysis data used for the DNN model training and validation. Section 3 gives the details of AOD detection modeling based on the DNN method. The validation of DNN-estimated AHI AODs against AERONET AOD measurements over China will be shown in Section 4. Section 5 is a summary and conclusions.

## 2. Study Area and Data Sources

### 2.1. Study Area

The study area (Figure 1) is the land part of China. Since China covers a huge territory with significant regional differences, the characteristics of air quality, terrain, landcover, population density and economic growth vary largely in different regions. Due to the influence of turbulent vertical stirring on aerosol dilution and chemical processes and meteorological processes in China, it is also significant on the temporal and spatial variability. All above-mentioned reasons can jointly lead a more complex spatiotemporal distribution of aerosol pollution for the entire area of China.

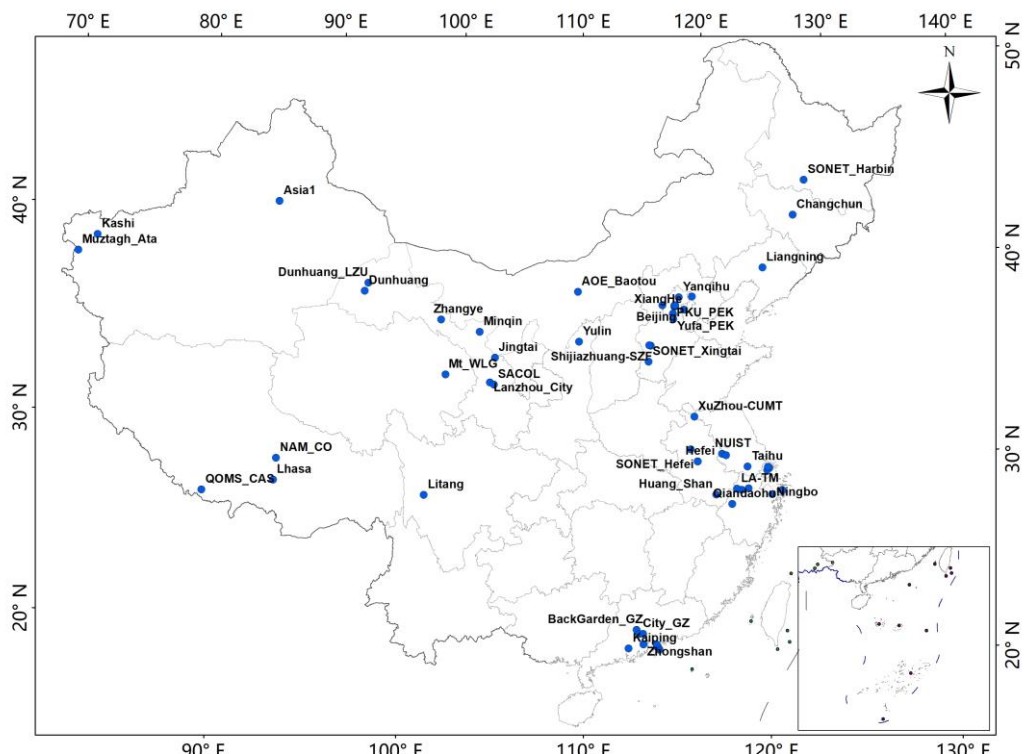

**Figure 1.** Study area and location of AERONET sites.

### 2.2. Datasets

#### 2.2.1. Himawari-8/AHI Data

The Himawari-8/AHI managed by JAXA has been in operation since 7 July 2015. Compared with similar sensors onboard the previous generation of geostationary satellites, the AHI was greatly improved. Its temporal resolution of full disk coverage is 10 min and that of dense coverage such as the Japan area is 2.5 min. The AHI is a 16-band multi-spectral sensor with a 500-m red band (0.64 μm) and 1-km green, blue and near-infrared bands (0.51, 0.47 and 0.86 μm). The spatial resolution of the remaining 12 bands ranging from 1.6 to 13.3 μm is 2 km. Increasing the spectral range of visible and near-infrared bands makes the aerosol retrieval over land possible from geostationary satellite data.

Three Himawari-8/AHI products were used in this study. One is the Himawari-8 level 1 band data, with a temporal resolution of 10 min and a spatial resolution of 5 km × 5 km. Its embedded dataset includes visible and near-infrared band albedo, mid-infrared band brightness temperature, view zenith angle (*vza*), view azimuth angle (*vaa*), solar zenith angle (*sza*), solar azimuth angle (*saa*), observation time, etc. Himawari-8/AHI level 3 hourly aerosol product provides AOD at 500 nm, Ångström exponent, AOD uncertainty and QA flag at a spatial resolution of 5 km. Himawari-8/AHI level 2 cloud product includes cloud optical thickness, cloud effective radius, cloud top temperature, cloud top height and cloud types (ISCCP definition) with a spatial resolution of 5 km and a temporal resolution of 10 min. Here, level 1 band data was mainly used for AOD inversion, and level 2 cloud and level 3 aerosol products were applied for cloud screening and AOD evaluation, respectively. All these datasets in the NetCDF format can be available from the website (https://www.eorc.jaxa.jp/ptree, accessed on 16 April 2022) published by JAXA.

#### 2.2.2. ECMWF Reanalysis v5 (ERA5) Data

The ERA5 reanalysis dataset has been widely used in regional environmental monitoring [49]. It is the latest generation of atmospheric reanalysis data released by the European Center for Medium-Range Weather Forecasts (ECMWF). The dataset is in the form of global grid with a conventional grid resolution of 0.25° × 0.25°. Compared with previous

generation products, the temporal resolution is increased from 6 h to 1 h, which makes it possible to study the diurnal variation of the troposphere. Additionally, the accuracy of ERA5 is significantly improved compared with the previous generation products [50]. ERA5 hourly and monthly reanalysis products contain different data subgroups, such as temperature, surface pressure and wind and soil data groups. In this study, hourly meteorological parameters, including temperature, air pressure and wind speed data, were extracted from the ERA5 dataset for DNN model training and AOD retrieving.

### 2.2.3. AERONET AOD Data

AERONET is a federation of ground-based remote sensing aerosol networks established and maintained by National Aeronautics and Space Administration (NASA) and PHOtométrie pour le Traitement Opérationnel de Normalisation Satellitaire (PHOTONS). It covers hundreds of sites around the world and provides low uncertainty and standardized data processing. The estimation accuracy of AERONET AOD product is ±0.02 [51]. Due to the high quality of the measurements, AERONET AOD observations are often taken as "basic facts" to validate satellite-based AOD retrievals. AERONET AOD data (available on the website https://aeronet.gsfc.nasa.gov/, accessed on 16 April 2022) offers three options for different quality levels: 1.0 (no filtering), 1.5 (cloud-screened and quality control) and 2.0 (quality-assured). Both Level 2.0 and Level 1.5 data were accessed for the validation of retrieved AHI AOD results in this paper.

### 2.2.4. Terra/MODIS and Aqua/MODIS AOD Data

For more than 20 years, Terra/MODIS and Aqua/MODIS have delivered stunning imagery of Earth and critical atmospheric parameters. MODIS AOD products are the most widely used aerosol dataset on both regional and global scales. There are four standard MODIS AOD products over land retrieved by the DT algorithm, the DB algorithm, the DBT algorithm combining DT and DB algorithms and the Multiangle Implementation of Atmospheric Correction (MAIAC) algorithm [52], respectively. MAIAC is a recent aerosol algorithm developed to retrieve AOD over land, which assumes that the space is consistent and stable [53,54]. Considering the influence of ground bidirectional reflectance, this algorithm adopts the time series data to dynamically separate aerosol and land contributions. Among all the standard MODIS AOD products, both the spatial resolution (1 km) and the data coverage rate of MAIAC AOD product are the highest, and the accuracy is also generally the highest in most regions. Therefore, the MAIAC AOD product (MCD19A2) was selected to evaluate the spatial distribution of the AHI AOD retrievals at the corresponding overpass times of the Terra and Aqua satellites.

## 3. Model and Methodology

The procedure of Himawari-8/AHI AOD retrieval based on a DNN model in this study is shown in Figure 2. Firstly, the AERONET AODs, ERA5 meteorological parameters, AHI multi-band data and AHI cloud mask in 2018 and 2019 needed to be matched with each other in different places and at different times. After cloud screening, quality control and normalization, the matched samples were randomly grouped into training (90%) and validation (10%) datasets. Feature selection was then conducted, and only the remaining features were used for DNN model training. Through error function evaluation, full connection network parameters setting, activation function selection and drop processing, the final model of AHI AOD estimation would be established with multiple iterations. Finally, AHI AOD in 2020 over China would be predicted, and the performance of our model would be evaluated by ground-based AERONET, MODIS and AHI AOD products.

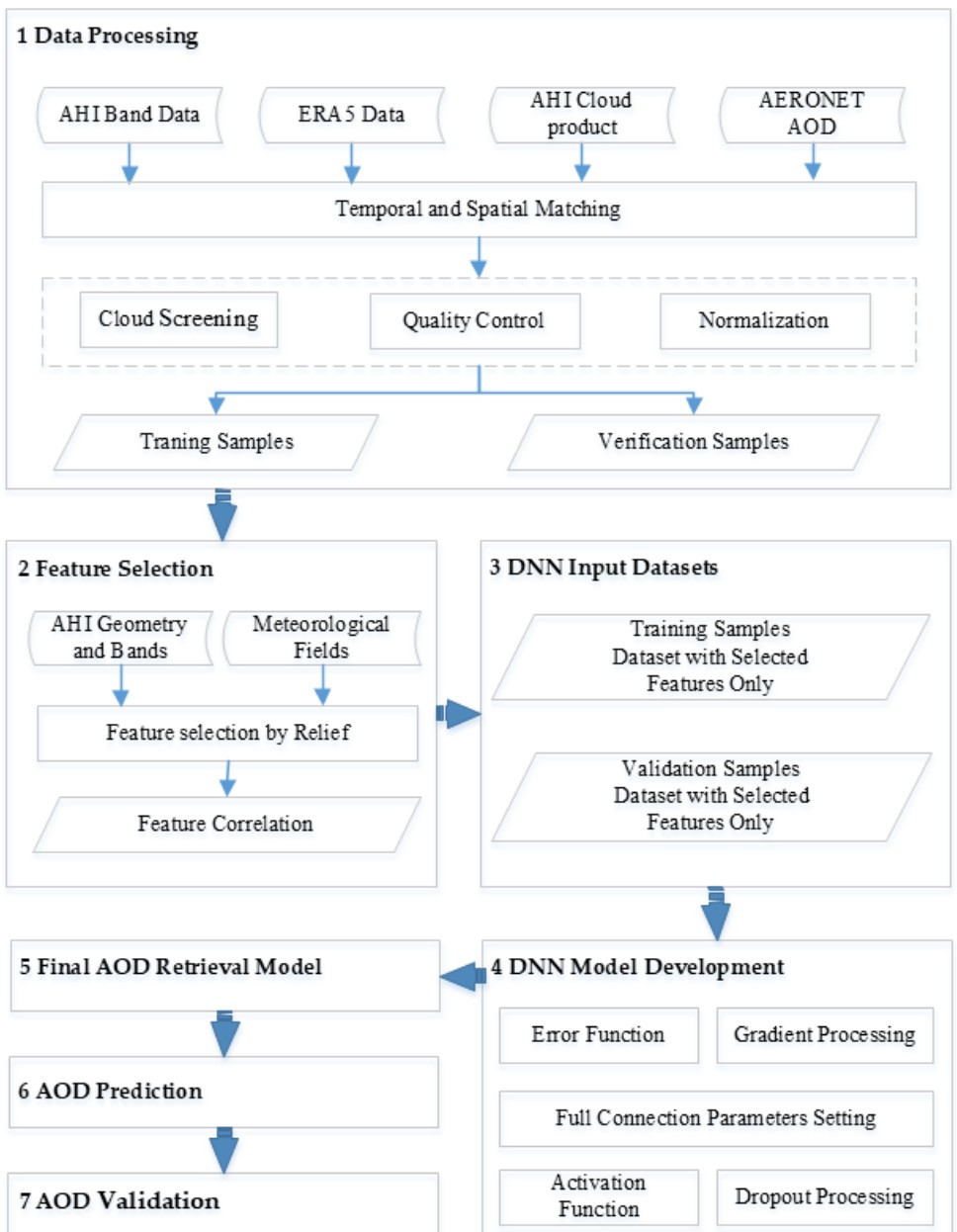

**Figure 2.** Flow chart of DNN model training and validation for Himawari-8/AHI AOD estimation.

### 3.1. Data Preprocessing

As shown in Table 1, the AERONET AOD product can provide AODs at 22 wavelengths ranging from 340 to 1640 nm. The MODIS AOD product involves AODs at wavelengths of 550 and 470 nm, and the AHI AOD product can only provide AODs at 500 nm. Therefore, in order to satisfy the requirement of comparison, AODs at 500 nm was extracted from the AERONET and AHI AOD products, and the MODIS AOD product was converted from 550 nm to 500 nm. Here, MODIS AOD at 500 nm was calculated using the Ångström exponent and AOD at 550 nm and 470 nm, given by

$$\partial_{470-550} = -\frac{\ln(\tau_{470}/\tau_{550})}{\ln\left(\frac{470}{550}\right)} \tag{1}$$

$$\tau_{500} = \tau_{550}\left(\frac{500}{550}\right)^{-\partial_{470-550}} \tag{2}$$



where $\tau_{470}$ and $\tau_{550}$ are MODIS AODs at 470 nm and 550 nm, respectively, $\partial_{470-550}$ is the Ångström index at 470–550 nm and $\tau_{500}$ is the result of AOD at 500 nm.

**Table 1.** List of AOD products at different wavelengths.

| AOD Product | Bands (nm) |
|---|---|
| AERONET AOD | 1640, 1020, 870, 865, 779, 675, 667, 620, 560, 555, 551, 532, 531, 510, 500, 490, 443, 440, 412, 400, 380, 340 |
| MODIS AOD | 470, 550 |
| Himawari-8 AOD | 500 |

Due to the relative homogeneous aerosol properties within a certain time–space boundary, it is necessary to select a proper spatiotemporal window for averaging and matching satellite band data and AERONET AOD observations. In this study, a time window with 15 min centered at the Himawari-8 observation time was used for averaging the AERONET AOD measurements. Due to the different spatial resolutions of AHI multi-band data, as shown in Table 2, spatial windows with 1, 2 and 4 pixels centered at each AERONET site were selected for AHI bands with a spatial resolution of 2 km, 1 km and 500 m, respectively.

**Table 2.** Bands and spatial resolutions of Himawari-8/AHI.

| Band | Wavelength (µm) | Spatial Resolution (km) | Band | Wavelength (µm) | Spatial Resolution (km) |
|---|---|---|---|---|---|
| 1 | 0.46 | 1 | 9 | 7.0 | 2 |
| 2 | 0.51 | 1 | 10 | 7.3 | 2 |
| 3 | 0.64 | 0.5 | 11 | 8.6 | 2 |
| 4 | 0.86 | 1 | 12 | 9.6 | 2 |
| 5 | 1.2 | 2 | 13 | 10.4 | 2 |
| 6 | 2.3 | 2 | 14 | 11.2 | 2 |
| 7 | 3.9 | 2 | 15 | 12.3 | 2 |
| 8 | 6.2 | 2 | 16 | 13.3 | 2 |

### 3.2. Feature Selection

Feature selection is one of important steps in building a ML model. It aims to reduce the input features to ML models by using only relevant features and to get rid of noise in the dataset. By selecting a subset of relevant features used in the AHI AOD model construction, the accuracy and complexity of our model would be affected directly. The filtering method is commonly used in the feature selection. Regardless of the model characteristics, it evaluates the significance of the explanatory variables. These methods are characterized by high computational efficiency and large simulation resistance. A Pearson Correlation Coefficient (PCC) is one of the typical filters. The correlation ranking considers the linear relationship between each predictor $X_j$ and the response variable $Y$:

$$\text{PCC(j)} = \frac{cov(X_j, Y)}{\sqrt{var(X_j) \times var(Y)}} \tag{3}$$

where *cov* and *var* represent the covariance and variance, respectively.

According to the radiative transfer theory, the spectral TOA signal finally received by a satellite sensor is related to sun-sensor view geometry, atmospheric states, aerosol types, AOD, sensor spectral characteristics and surface types. Therefore, all parameters mentioned above must be involved in the forward simulation for satellite-based AOD retrieval by physical algorithms. In this study, there is no need to assume the atmospheric states, aerosol modes and surface types in advance, because the forward radiative transfer simulation is

not required by AHI AOD modeling based on DNN. Therefore, only the angles related to sun-sensor geometry, AHI TOA reflectance and bright temperature at bands 1~16 were selected as candidate features. Additionally, aerosol distribution is closely associated with meteorological conditions and landcover types [55,56]. Here, surface pressure (*sp*), the temperature at 2 m (*t2m*) and wind speed (*ws*) values extracted from ERA5 reanalysis dataset were used to characterize meteorological conditions. The Normalized Difference Vegetation Index (NDVI) values calculated from Himawari-8/AHI data were employed to simply describe the spatiotemporal characteristics of landcover types.

The correlations between AOD and all candidate features were studied by a correlation analysis experiment as shown in Figure 3, with PCC as the selected filter. It is indicated that importance values of AHI TOA measurements for bands 1~6 (*bd1, bd2, bd3, bd4, bd5* and *bd6*) are higher than those for bands 7~16, which is consistent with the study of Chen et al. [46]. The sun-sensor view geometry angles (*sza, saa, vza* and *vaa*); meteorological parameters (*sp, t2m* and *ws*) and NDVI are also shown a relative high importance. Therefore, *bd1, bd2, bd3, bd4, bd5, bd6, sza, saa, vza, vaa, sp, t2m, ws* and the NDVI were retained as input features for our AOD retrieval model based on DNN.

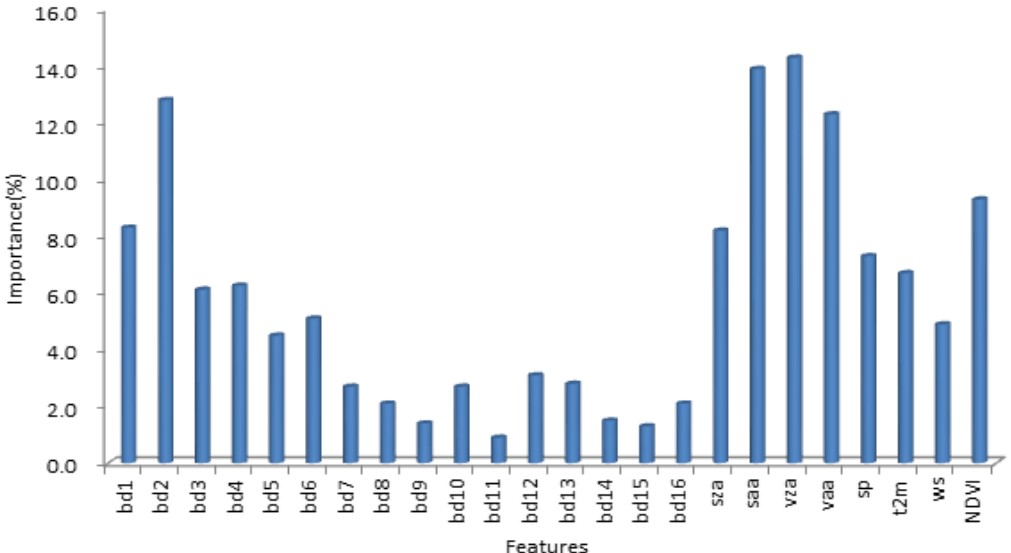

**Figure 3.** Himawari-8/AHI aerosol retrieval importance of input features calculated by the PCC filter.

### 3.3. DNN Model

As one of the most typical depth estimation and image recognition applications, DNN usually consists of three parts: input plane, hidden layers and output layer. The layer of the DNN model is composed of independent neurons, which only interact with adjacent layers. Interactions include two operation processes: prefeeding and back propagation [57].

In the prefeeding stage, the data is transferred between two adjacent layers through the combination of linear function and nonlinear activation function, as shown below:

$$a_l = \sigma(W_l \cdot a_{l-1} + b_l) \tag{4}$$

where $l$ is the layer number, $a$ is the output vector of a layer, $W$ is the weight matrix, $b$ is the offset vector and $\sigma(.)$ is the activation function as shown in Equation (5):

$$\sigma(x) = \begin{cases} x, & if\ x \geq 0 \\ 0, & if\ x < 0 \end{cases} \tag{5}$$

Inverse multiplication is the process of updating the weights and deviations found in previous calculations. Firstly, the loss function is defined by the root mean square error function. Then, the offset and weight are gradually optimized by solving the gradients

of the loss function. The preparation process of establishing the DNN model includes iterations of the gradient descent mentioned above. The Adam optimization algorithm [58] is used for completing gradient degradation and weight initialization. The weights in the iteration are updated as follows:

$$W_{l,t+1} = W_{l,t} - \alpha \cdot f(\nabla Loss_l, \beta_1, \beta_2, t) \tag{6}$$

where $\alpha$ is the initial learning rate ($\alpha$ = 0.01), $\beta_1$ and $\beta_2$ are the initial exponential decay rates ($\beta_1$ = 0.900 and $\beta_2$ = 0.99), $\nabla Loss$ is the gradients loss and $f(.)$ is the function with respect to the above parameters and the iteration number $t$ [58]. The offsets are updated in the same way.

The DNN model used for the AHI AOD retrieval (Figure 4) was established with the input layer consisting of features selected in Section 3.2 and the outer layer consisting of AODs at 500 nm. The model was set with four hidden layers, each of which is fully connected and needs to be activated by the ReLU function to realize nonlinearity and drop some nodes to prevent overfitting. The AERONET AOD was defined as the model tab for training and validation.

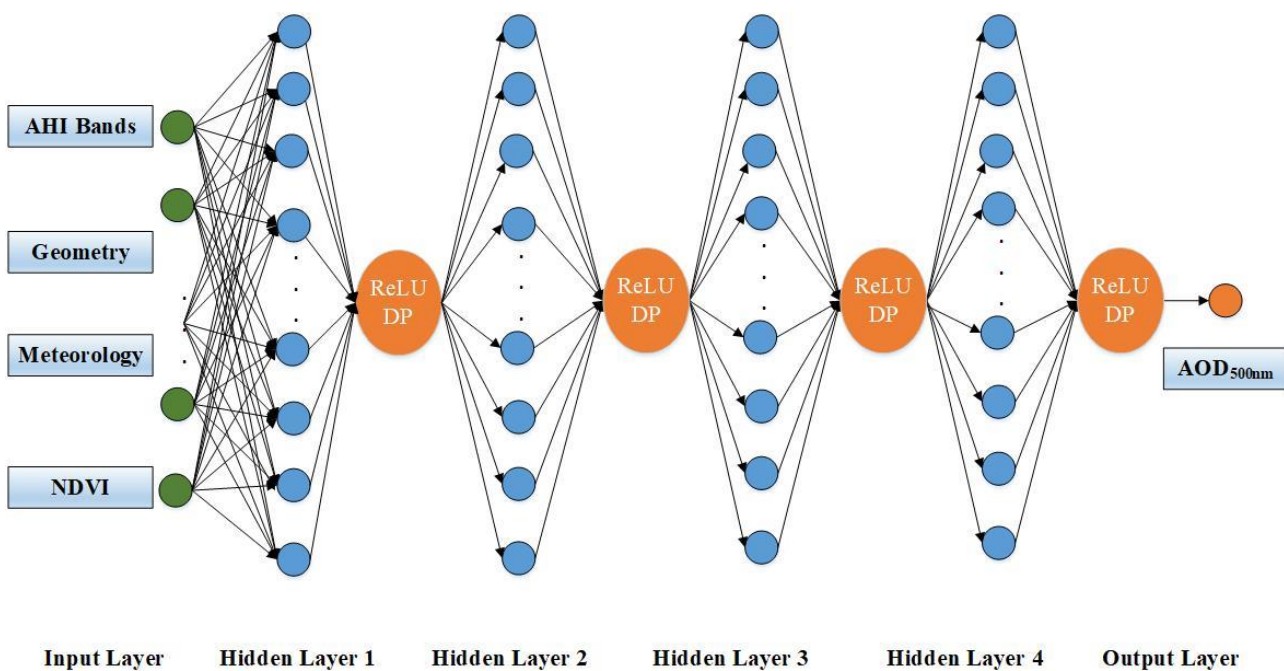

**Figure 4.** The architecture of the DNN model for Himawari-8/AHI aerosol retrieval (input nodes: *bd1, bd2, bd3, bd4, bd5, bd6, sza, saa, vza, vaa, sp, t2m, ws* and the NDVI; output node: Himawari-8/AHI AOD at 500 nm; DP: Drop Layer and ReLU: Rectified Linear Unit Layer).

### 3.4. Hyperparameter Tuning and Model Validation

The goal of hyperparameter tuning is to discover the best hyperparameter combination that minimizes a preset loss function and produces better results. Grid search was used to obtain the local optimal solution of network hyperparameters, including minimum batch, learning speed, network depth and activation function. In the DNN model, the sample data was randomly divided into two parts: 10% of the samples for validation, the remaining 90% samples for model training. $R^2$ and *RMSE* values of training and validation were taken as evaluation criteria to assess the model performance. The final DNN model would be determined through hyperparameter tuning by changing some key parameters. When the DNN model of AOD detection trained by datasets in the year of 2018 and 2019 was stable, it was used to predict AHI AODs in 2020. Here, ground-based AERONET measurements as truths were adopted to comprehensively validate the accuracy of retrieved AOD results.

AHI and MODIS AOD products were mainly used to evaluate the reasonability of our results in the spatial distribution.

## 4. Results and Discussion

### 4.1. Overall and Hourly Validation on Sites

Figure 5 shows the scatterplots for our DNN-estimated AHI AOD at 500 nm against the ground-based AERONET measurements in 2020. The statistics were fitted by 59,716 matchups from retrieved AOD results at AERONET sites in China. A good correlation between them is shown with a high $R^2$ of 0.8168 and a low RMSE of 0.1546. Since the learning sample data were generated randomly from the training sample dataset, the prediction stability of the trained models was disturbed to some extent. According to the statistics of 100 random disturbances of learning samples, the $R^2$ varies from 0.7915 to 0.8229 with the RMSE ranged of 0.1528~0.2037. The narrow variation ranges of both $R^2$ and *RMSE* mean that the prediction capabilities of these models are very similar. The result given in Figure 5 is a case of these models that can represent the average capability of AOD retrieval, and all the subsequent comparisons were conducted based on this model.

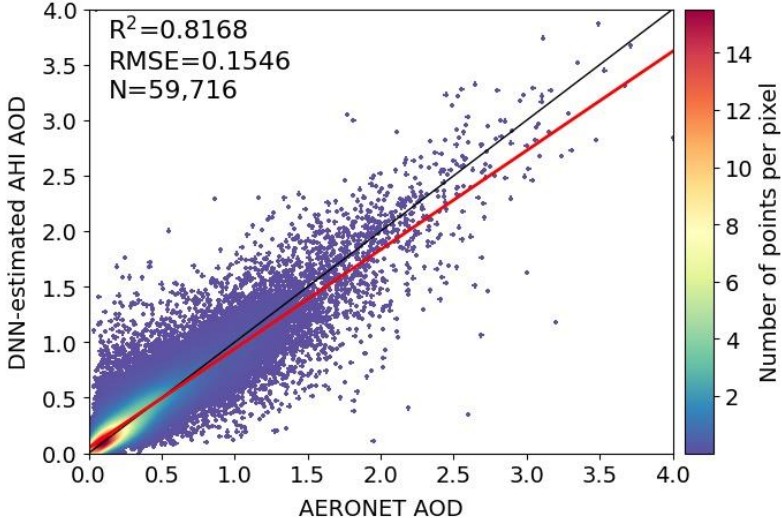

**Figure 5.** Comparison between DNN-estimated AHIAOD and AERONET AOD.

To compare the accuracies of retrieved AOD by the DNN model with the AHI AOD product, both were validated by AERONET AOD measurements over China in 2020, respectively. There are 24,270 retrieved AODs matched with both AERONET and AHI AOD products. As shown in Figure 6, both DNN-estimated AHI AOD and AHI AOD product are underestimated, but DNN-estimated AHI AOD has a better agreement with AERONET AOD than that from AHI product. $R^2$ reaches up to 0.8702 for our algorithm with a small *RMSE* of 0.1326, while $R^2$ is only 0.4869 for the AHI AOD product with a large *RMSE* of 0.2917 in China. Compared with the AHI product, the algorithm used in this study effectively improved the AOD retrievals in the aspects of both accuracy and data coverage.

Hourly validations of AODs retrieved by the DNN model and AHI AOD product are shown in Figure 7. Since the AERONET AOD measurements at 0:00 UTC are missing, hourly validations in the daytime were conducted for 1:00~12:00 and 22:00~23:00. It should be noted that all the timestamps mentioned in this paper are UTC time, not local time. Each subfigure in Figure 7 corresponds to one hour on the time, where the estimated AOD (red dot) and AHI AOD product (green dot) at the same time were compared, and the AERONET AOD was treated as the true value. For all sets of hourly validation, $R^2$ values of AODs detected by our algorithm (0.454~0.969) were much higher than those of the AHI

AOD product (0.278~0.864), and *RMSEs* (0.03~0.173) were lower than those of AHI AOD product (0.140~0.440).

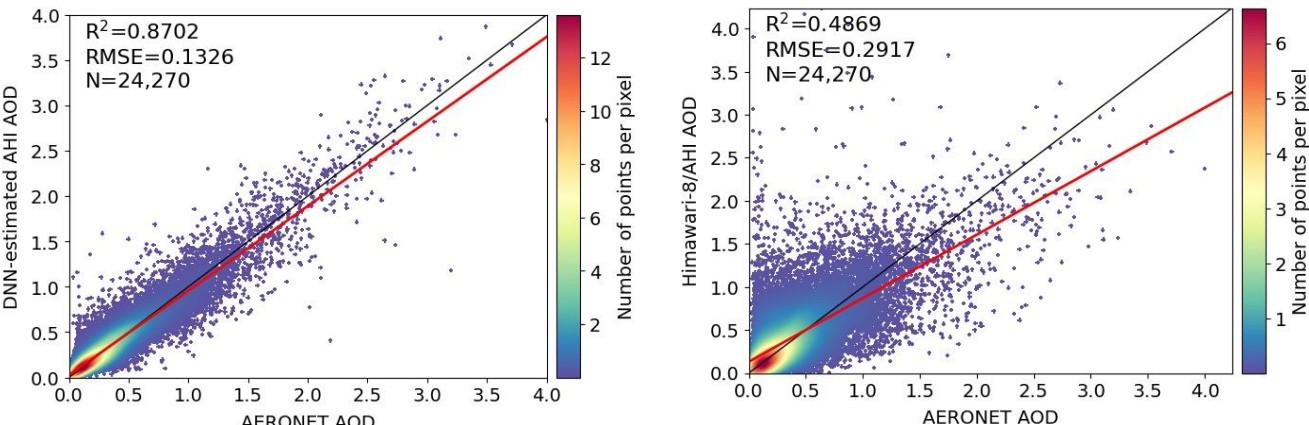

**Figure 6.** Comparison between the AHI AOD and AERONET AOD ((**left panel**): DNN-estimated AHI AOD; (**right panel**): AHI AOD product).

Especially, due to the large solar zenith angle in the early morning (22:00 and 23:00) and in the late afternoon (11:00 and 12:00), the path length of lights passing through the atmosphere is long, which makes the AOD inversion very difficult based on the physical methods. However, the AOD retrieval algorithm based on DNN model could well overcome this problem to some extent from Figure 7A,B,M,N. For the validation at 23:00, $R^2$ increases from 0.278 for the AHI product to 0.777 for AOD retrievals, while *RMSE* correspondingly decreases from 0.44 to 0.103. For the validation at 11:00, the performance of our algorithm based on DNN model with $R^2 = 0.454$ and *RMSE* = 0.173) is also better than AHI product with $R^2 = 0.343$ and *RMSE* = 0.242. Here, due to the limitation of the amount of valid AHI TOA reflectance, the available AOD retrievals are very few for comparison at 22:00 and 12:00.

### 4.2. Evaluation of Spatial Distribution on a Regional Scale

AODs of Terra/MODIS, Aqua/MODIS and Himawari-8/AHI products were chosen to evaluate the reasonability of spatial distribution for hourly AHI AOD retrievals by the DNN model. On a regional scale, a heavy air pollution (haze) event occurred in eastern China on 8 November 2020. Hourly AHI true-color RGB images, AHI AOD retrievals by DNN model, AHI AOD products and MODIS AOD products over eastern China at 1:00~7:00 are correspondingly depicted in Figure 8. Similar to Figure 8, Figure 9 gives the spatial distributions of AOD at 2:00 but over China. Areas with high or low AOD values for our AOD retrievals are generally consistent with those for both AHI and MODIS AOD products. Based on the comparisons at 2:00 and 5:00, the spatial distribution of our AOD retrievals has a better agreement with MODIS data than AHI. Moreover, it needs to be noted that the data coverage rate of DNN retrieved AOD is also higher than that of AHI AOD products, because AOD cannot be well-retrieved over the bright targets, and many pixels loaded by heavy aerosol are always misidentified as cloud pixels in the JAXA AOD retrieval algorithm. Normally, the reliability of the MODIS AOD product is higher than other AOD products, and the accuracy of AOD retrievals from a sun-synchronous satellite is greater than that from a geostationary satellite. Therefore, it means that DNN-estimated AHI AOD can reasonably reflect air pollution regions with high AOD levels and can accurately capture hourly variations of aerosol loadings on a regional scale.

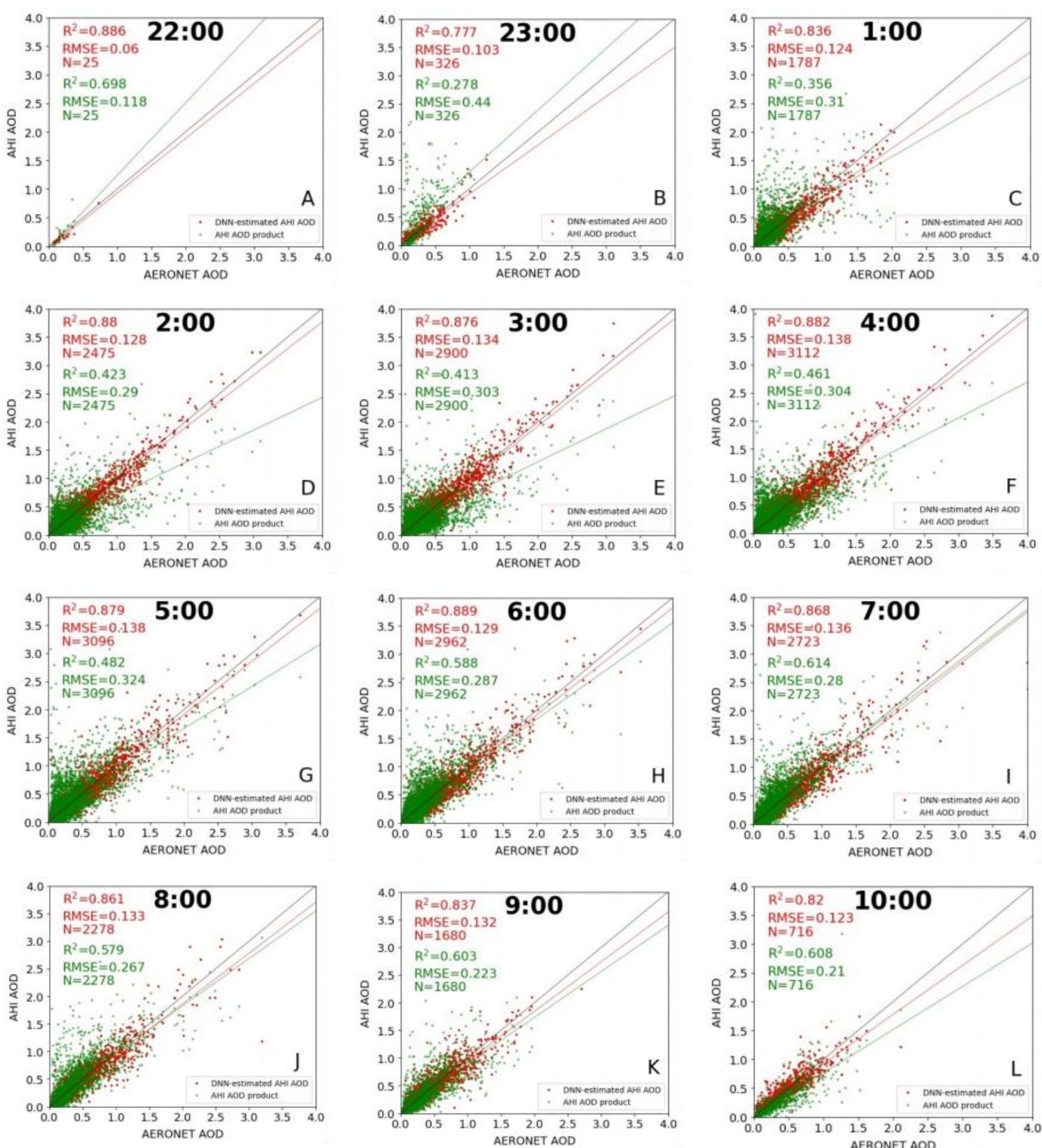

**Figure 7.** *Cont.*

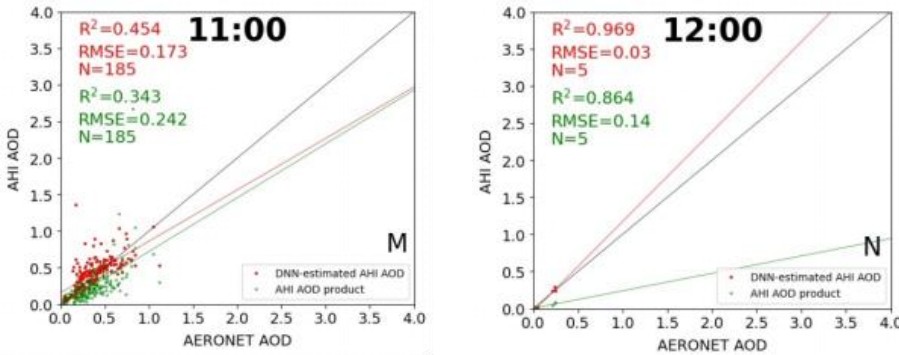

**Figure 7.** Hourly comparisons between the AHI AOD and AERONET AOD (red dot: DNN-estimated AHI AOD; green dot: AOD extracted from AHI product; panels (**A**–**N**): 22:00~23:00 and 1:00~12:00 (UTC)).

01:00 on 8 November 2020

02:00 on 8 November 2020

03:00 on 8 November 2020

**Figure 8.** *Cont*.

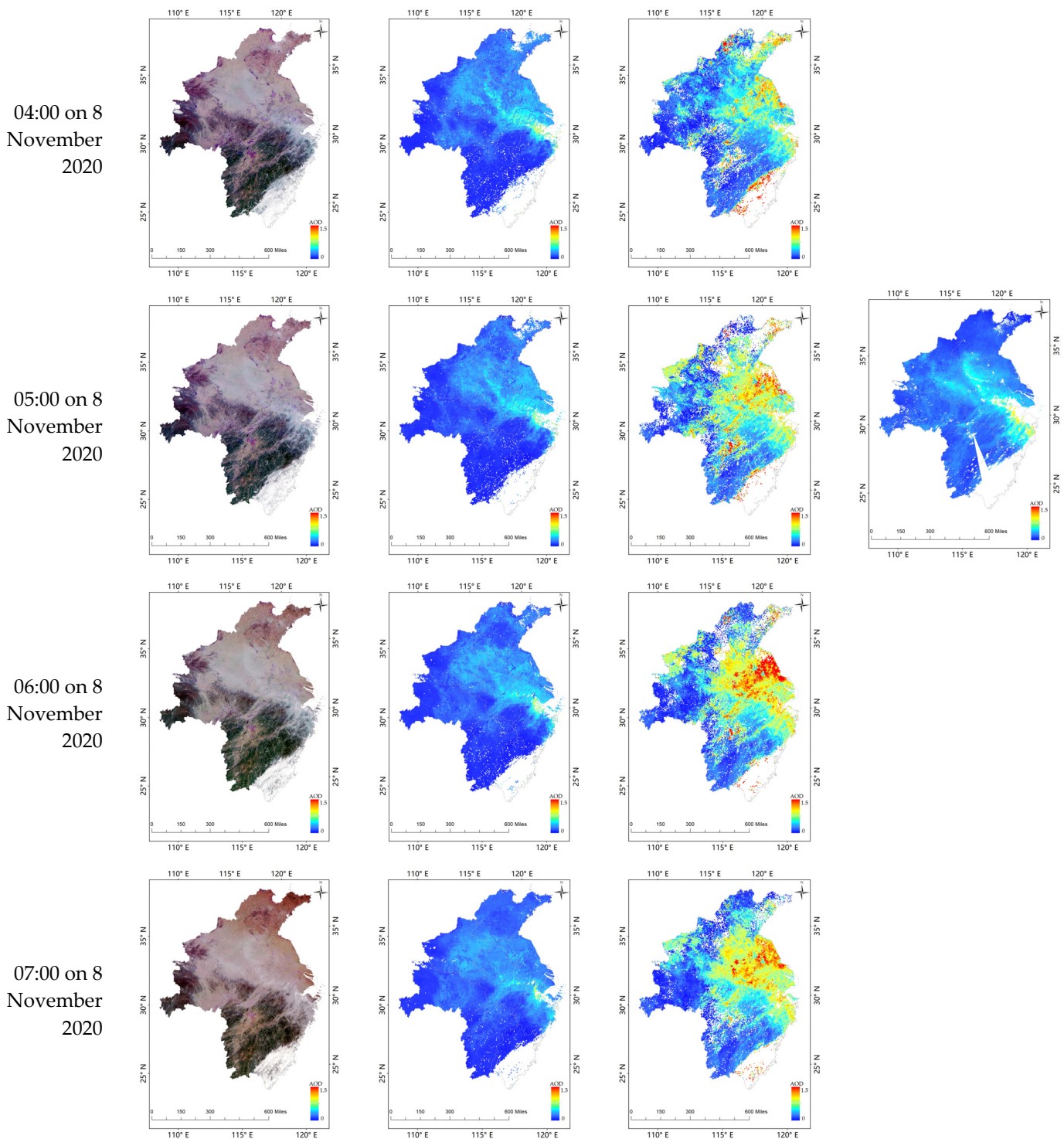

**Figure 8.** Hourly AHI true-color RGB images (1st column of panels), DNN-estimated AHI AOD (2nd column of panels), AHI AOD products (3rd column of panels) and MODIS AOD products (4th column of panels) over Eastern China at 1:00~7:00 (top to bottom) on 8 November 2020.

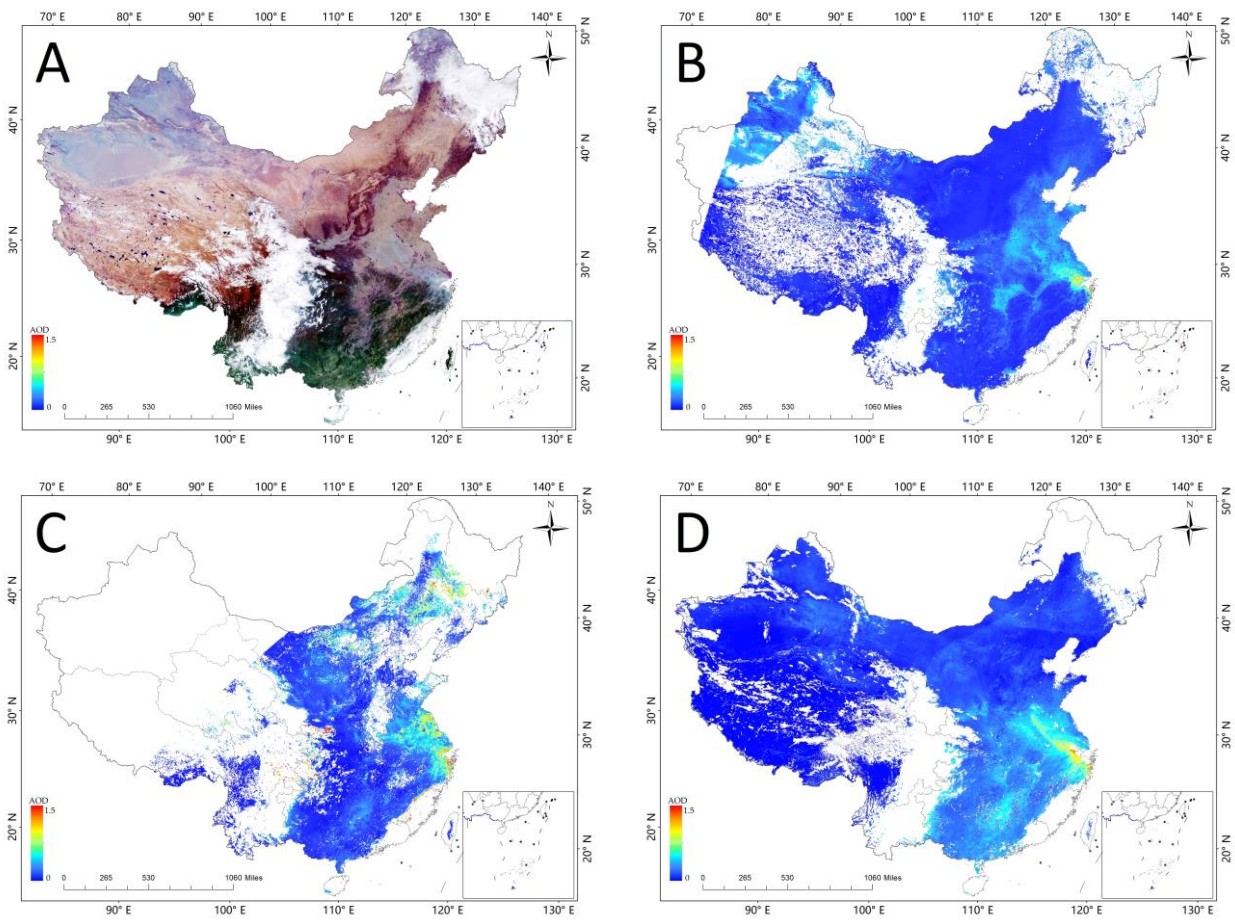

**Figure 9.** AHI true-color RGB images (**A**), DNN-estimated AHI AOD (**B**), AHI AOD product (**C**) and MODIS AOD product (**D**) over China at 2:00 on 8 November 2020.

### 4.3. Seasonal Modeling and Validation

Since the aerosol types and meteorological parameters largely vary with the season, we tried to investigate whether seasonal DNN modeling can further improve the accuracy of AHI AOD retrievals. Following the modeling procedure in Section 3, the sample dataset created from the data of 2018 and 2019 was divided into four parts by season, and the seasonal DNN models were trained by seasonal sample datasets. Here, Models 1~4 correspond to the established models of spring, summer, autumn and winter, respectively. Model 5 trained by all samples was applied for contrast, which is the same as the DNN model used in Figure 5. The AHI AODs in 2020 were separately calculated by Models 1~5.

Seasonal validation results of AHI AOD estimations against AERONET AOD measurements are given in Figure 10 and Table 3. Although seasonal Models 1~4 could well-predict AOD in the spring, summer, autumn and winter, respectively, any of them was not suitable to detect the AOD of the other three seasons. For spring, the $R^2$ of AOD estimated by Model 1 reaches 0.8199, but the $R^2$ of Model 2 is only 0.4285. For the winter, the performance in AOD detection is the best for Model 4 with $R^2 = 0.8508$ and $RMSE = 0.1347$ but the worst for Model 2 with $R^2 = 0.3575$ and $RMSE = 0.3476$. Compared with the seasonal models, Model 5 has a better applicability for AOD retrieval in different seasons. Especially, Model 5 performed best among all the five models in the spring and summer. Therefore, a separate DNN model built by season can hardly increase the accuracy of the AOD retrieval to a certain extent. Generally, due to the good learning ability of DNN, the larger number of samples the DDN learns from, the better performance of AOD prediction the established model shows. Unlike the physical AOD retrieval algorithm, which adopts different in-

version strategies to solve the uncertainties introduced by surface and aerosol type, a ML algorithm such as DNN has better universality and robustness.

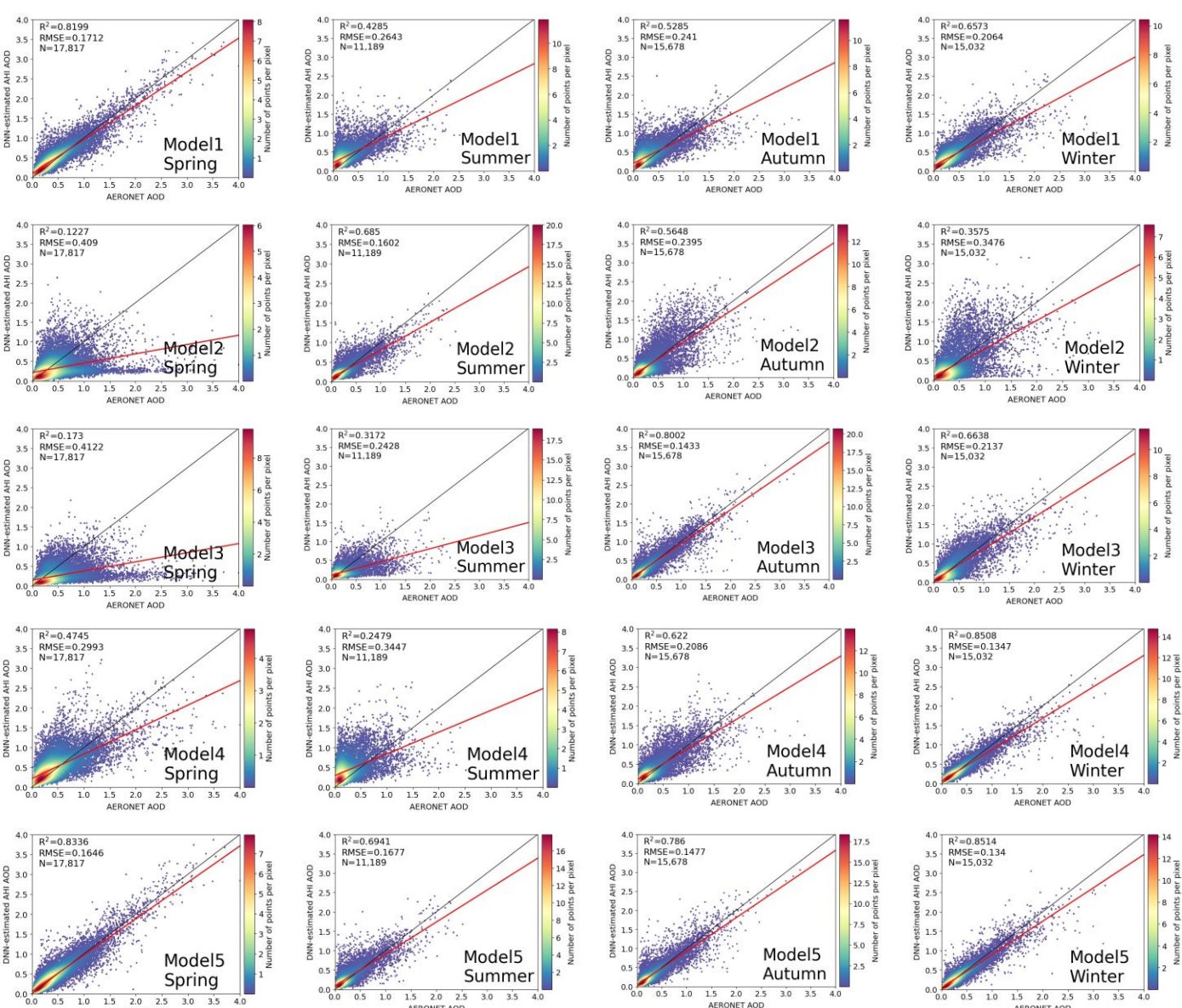

**Figure 10.** Seasonal comparisons between DNN-estimated AHI AOD and AERONET AOD for Models 1~5 (top to bottom rows). Left to right columns are corresponding to spring, summer, autumn and winter, respectively.

**Table 3.** $R^2$ and *RMSE* list of seasonal comparisons shown in Figure 10.

|  | Model 1 $R^2$ | Model 2 $R^2$ | Model 3 $R^2$ | Model 4 $R^2$ | Model 5 $R^2$ | Model 1 *RMSE* | Model 2 *RMSE* | Model 3 *RMSE* | Model 4 *RMSE* | Model 5 *RMSE* |
|---|---|---|---|---|---|---|---|---|---|---|
| Spring | 0.8199 | 0.1227 | 0.1730 | 0.4745 | 0.8336 | 0.1712 | 0.4090 | 0.4122 | 0.2993 | 0.1646 |
| Summer | 0.4285 | 0.6850 | 0.3172 | 0.2479 | 0.6941 | 0.2643 | 0.1602 | 0.2428 | 0.3447 | 0.1677 |
| Autumn | 0.5285 | 0.5648 | 0.8002 | 0.6220 | 0.7860 | 0.2410 | 0.2395 | 0.1433 | 0.2086 | 0.1477 |
| Winter | 0.6573 | 0.3575 | 0.6638 | 0.8508 | 0.8514 | 0.2064 | 0.3476 | 0.2137 | 0.1347 | 0.1340 |

### 4.4. Impact of Meteorological Features

Aerosol mainly exists in the lower atmosphere. The occurrence of heavy aerosol pollution events is not only strongly affected by excessive emission of air pollutants, but is also closely associated with meteorological conditions. In addition to human activities, meteorological factors cannot be ignored. Meteorological factors have a significant impact on the aggregation, transmission, and diffusion of air pollutants, of which temperature, air pressure and wind speed are typical representatives. Therefore, in this study, besides AHI data, meteorological parameters as input features were involved in DNN modeling for AHI AOD detection.

To investigate the impact of meteorological factors on the DNN-estimated AHI AOD, eight cases with different input meteorological features (i.e., *sp*, *t2m* and *ws*) were tested and validated based on AERONET AOD measurements (Figure 11 and Table 4). Obviously, the retrieved AOD result of Case 1 trained by all three meteorological features is the best among Cases 1~8. With consideration of the meteorological features, the $R^2$ of DNN-estimated AHI AOD increases by 10.47%, and *RMSE* decreases by 14.82%. Compared with *t2m* and *ws*, *sp* has a greater influence on the improvement of AHI AOD estimation. For Case 5 trained by *sp*, a larger $R^2$ of 0.7947 and a smaller *RMSE* of 0.1615 was obtained in comparison to Cases 6, 7 or 4 trained by one or two features of *t2m* and *ws*. The $R^2$ of the DNN model decreases slightly by about 0.001~0.035 without regarding *t2m* or/and *ws*, while the $R^2$ decreases significantly by about 0.029~0.053 if *sp* is removed from selected features. The main reason may be that pressure patterns can objectively and accurately reflecting the evolution of aerosol pollution process and regional characteristics under most conditions [59,60]. The change of pressure reflects the change of the boundary layer height, and affects the diffusion capacity of the pollutants. Meanwhile, wind and temperature are also closely related to the spatial and temporal distribution of atmospheric aerosol by changing the dynamics of the air movement.

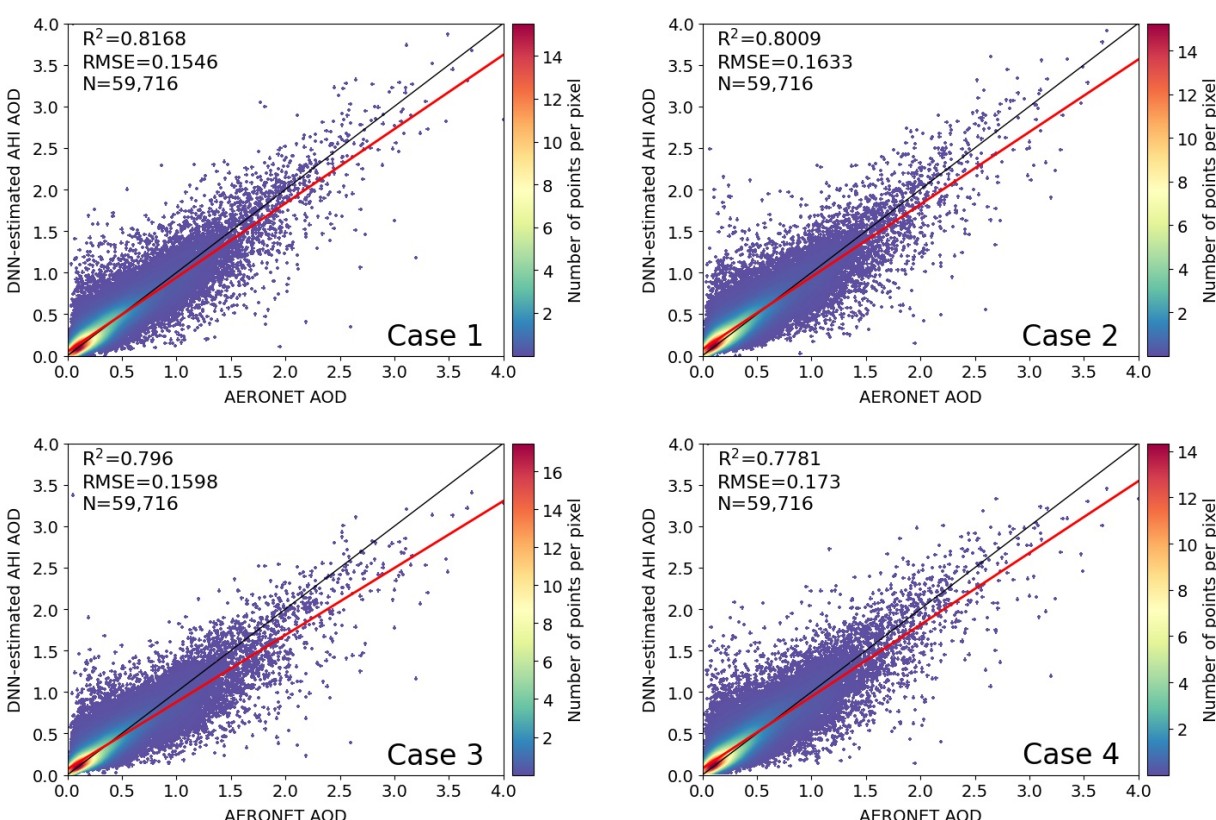

**Figure 11.** *Cont.*

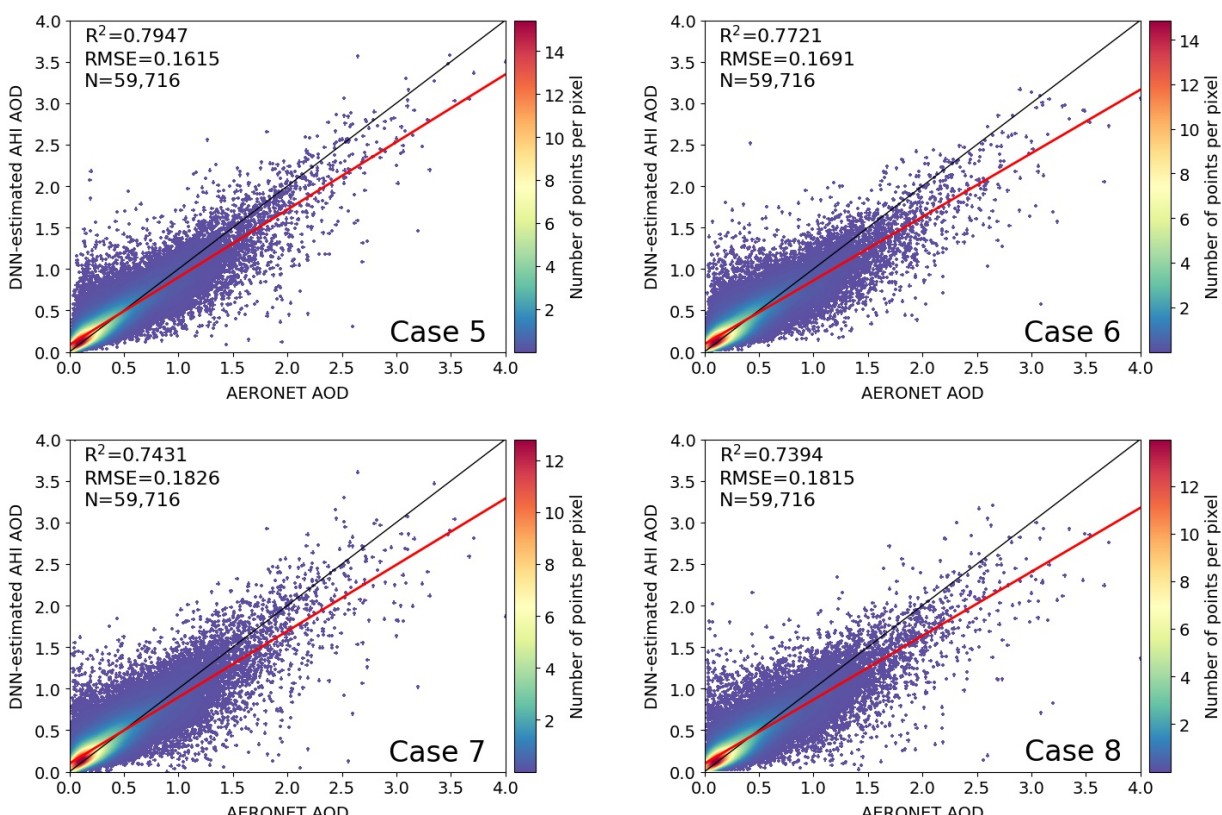

**Figure 11.** Comparisons between DNN-estimated AHI AOD and AERONET AOD for Cases 1~8 with different input meteorological features.

**Table 4.** $R^2$ and RMSE of 8 cases with different input meteorological features.

| Cases | Meteorological Factors Included | $R^2$ | *RMSE* |
|---|---|---|---|
| Case 1 | *sp, t2m, ws* | 0.8168 | 0.1546 |
| Case 2 | *sp, t2m* | 0.8009 | 0.1633 |
| Case 3 | *sp, ws* | 0.7960 | 0.1598 |
| Case 4 | *t2m, ws* | 0.7781 | 0.1730 |
| Case 5 | *sp* | 0.7947 | 0.1615 |
| Case 6 | *t2m* | 0.7721 | 0.1691 |
| Case 7 | *ws* | 0.7431 | 0.1826 |
| Case 8 | *none* | 0.7394 | 0.1815 |

## 5. Conclusions

A Himawari-8/AHI AOD detection algorithm was proposed based on the DNN model with the combination of satellite TOA reflectance, sun-sensor geometries, meteorological factors and NDVI data. Treated AERONET AOD measurements as truths, the DNN model was trained by the sample dataset in 2018 and 2019. AHI AODs at 500 nm over China in 2020 were calculated, and AOD retrieval results were compared with AERONET, MODIS and AHI AOD products. The following conclusions can be drawn:

(1) The DNN model has a great potential in hourly AOD retrieval for Himawari-8 satellite in China. Based on the validation against AERONET AOD, the accuracy of DNN-estimated AHI AOD with $R^2$ of 0.8702 and *RMSE* of 0.1326 is much higher than that of the AHI AOD product with $R^2$ of 0.4869 and *RMSE* of 0.2917. Especially, the DNN model can improve the performance of AHI AOD retrieval in the early morning and in the late afternoon.

(2) For the spatial distribution, DNN-estimated AHI AOD is more consistent with the MODIS AOD product than the AHI AOD product. Therefore, our AOD results can

detect air pollution regions with high AOD levels well and can accurately capture hourly variations of aerosol loadings on the regional scale.

(3) For the proposed algorithm framework, meteorological features (i.e., *sp*, *t2m* and *ws*) were involved into the DNN model training, which are not used in the physical AOD retrieval methods based on the atmospheric radiation transferring model. With consideration of the meteorological features, the $R^2$ of DNN-estimated AHI AOD increases by 10.47%, and the *RMSE* decreases by 14.82%. Among all the meteorological factors, *sp* has the greatest influence on the AHI AOD estimation.

With automatic feature learning and powerful generalization ability, ML methods make better use of the spatiotemporal AOD information and provide both higher accuracy and reliability in data mining. Data-driven AOD predictions based on machine learning models and multi-source data integration can promote the monitoring and modeling of the air quality over China.

**Author Contributions:** Conceptualization, M.F. and Y.C.; methodology, M.F., L.C. and Y.C.; software, Y.C.; validation, J.T., Z.L. and Z.W.; formal analysis, Y.C. and M.L.; investigation, M.L.; resources, J.T., Z.W. and Z.L.; data curation, Y.C.; writing—original draft preparation, Y.C.; writing—review and editing, Y.C.; visualization, Y.C.; supervision, M.F.; project administration, M.F. and funding acquisition, M.F., J.T. and L.C. All authors have read and agreed to the published version of the manuscript.

**Funding:** This research was funded by the National Natural Science Foundation of China (Grant No. 41830109 and No. 41871254) and Guangxi Key Research and Development Project (Grant No. Guike AB20238015).

**Data Availability Statement:** Data used in the reported studies were obtained from websites, as indicated in the text.

**Acknowledgments:** The authors acknowledge the JAXA, ECMWF (https://www.ecmwf.int/, accessed on 16 April 2022), the AERONET team (https://aeronet.gsfc.nasa.gov, accessed on 16 April 2022) and MODIS (https://modis-atmosphere.gsfc.nasa.gov/, accessed on 16 April 2022) groups for the satellite and ground-based data and their hard work. We would like to thank the reviewers for their constructive comments and suggestions.

**Conflicts of Interest:** The authors declare no conflict of interest. The funders had no role in the design of the study; in the collection, analyses or interpretation of the data; in the writing of the manuscript or in the decision to publish the results.

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
