# Peer review of "Himawari-8/AHI Aerosol Optical Depth Detection Based on Machine Learning Algorithm"

_remotesensing, doi:10.3390/rs14132967_

Round 1

Reviewer 1 Report

I found the paper interesting and well written.

I suggest only to include, in the introduction, how AOD can be of interest also for modelling air quality at ground level. As i.e. done in https://www.sciencedirect.com/science/article/pii/S1309104219300595, https://academic.oup.com/rpd/article/137/3-4/280/1620553, https://www.sciencedirect.com/science/article/pii/S2352938522000246.

This aspect can give more importance to the paper, as AOD-derived PM2.5 can be really useful, providing (through further transformation) a view of PM2.5 ground-level concentrations, also in locations where concentrations are not available.

Reviewer 3 Report

This manuscript should be rejected. Because I came to the conclusion that the authors did not make the effort to carefully read through the manuscript they submitted, I am not willing to share the 75 recommendations for specific improvements that I recorded while reading through the manuscript. Instead, I only point out some of the main issues and leave it to the authors to read the manuscript carefully and correct any additional issues they find (should they decide to resubmit the manuscript). 

1.

Several panels in the right-side column of Figure 8 are missing. The missing panels should have shown outputs from the presented algorithm.

2.

Figure 11 does not show anything about seasons or about the models mentioned in the figure caption and in the referring text (Lines 403-405). Instead, the figure is just a copy of Figure 6.

3. 

The Table 4 mentioned in Lines 403-406 does not exist. The manuscript does include a table called Table 4, but that table concerns different issues (as noted in Line 429) and has nothing to do with the seasons or models discussed in Lines 403-406. 

4.

At some places, the manuscript contains not actual text suitable for readers, but words that seem to be notes on what topics the manuscript should cover. For example, Line 279 contains the three-word sentence saying only “PCC filter type.” Another example is Lines 331-332, where the text says “Preparation, validation and testing of R2 and RMSE - statistical overview of the overall characteristics of the model”. 

5. 

The manuscript contains numerous small errors that would not be a big problem individually but become a concern because of their sheer number.  Some examples:
- In Figure 10, the results for Model 5 are shown in the top row and not in the bottom row, which contradicts the text in Lines 400-402. Also, the caption explains the columns twice, but does not explain the rows at all. 
- All acronyms should be explained right where they are first used, even if the explanation seems obvious. Although many (but not necessarily all) readers could probably figure out that DNN in Line 212 refers to “deep neural network”, many would not know what other unspecified acronyms (for example FSS in Line 252 or PCC in Line 254) mean.
- Line 229 contains some unexpected spirals, and Line 290 contains the peculiar word “Xf”.
- The Li et al. reference mentioned in Line 290 is not listed in the “Reference” section, as reference [46] is not by Li et al.
- Lines 301 and 302 should agree on capitalization. Since the equation in Line 301 contains lowercase letters for parameters l and a, these letters should not be capitalized when the parameters are explained in Line 302. On the other hand, since the equation capitalizes the parameter W, it should also be capitalized in Line 302. 
- Line 303 contains an empty pair of parentheses. I guess sigma from the line above should be moved inside. 
- The first letter of references to specific lines, tables, figures, equations, or sections should all be capitalized. For example, Line 222 should say “Table 1” instead of “table 1”. 
- Periods (dots) are missing at the end of some sentences, for example in Lines 296, 306, or 307.

6. 

The paper should provide sufficient details (for example, the time intervals considered “morning” and “evening” should be specified in Section 4.2) and physical insights (for example, Section 4.5 could discuss briefly the physical reasons why pressure, temperature, and wind can speed affect AOD predictions) and should discuss limitations and potential future uses (for example, Section 6 could discuss whether the presented method could work for other regions or satellites).

Round 2

Reviewer 2 Report

I have one doubt regarding Equation 2.

Can authors justify why they have used (470/550) on the right hand side of the equation. I am of the view that it should be (500/550). Can authors justify how they have derived this equation using Angstrom law and linear interpolation as mentioned in line 276-277?? 

Reviewer 3 Report

My original review stated that “Because I came to the conclusion that the authors did not make the effort to carefully read through the manuscript they submitted, I am not willing to share the 75 recommendations for specific improvements that I recorded while reading through the manuscript” and included a recommendation to reject the manuscript. 

Although the manuscript was not rejected, I believe my review at least made the point that the time and effort of volunteer reviewers should be valued and respected, and that authors should check their manuscripts before submission instead of leaving reviewers with basic quality-control tasks that the authors themselves are supposed to do. With this in mind—and in the interest of readers—I relent, and I am now willing to share my comments on what improvements the manuscript needs to become suitable for publication.  

First off, the authors adequately addressed most of the issues highlighted in my original review, with the following three exceptions:

1.

My original review included the comment that the time intervals considered “morning” and “evening” should be specified in Section 4.2. The authors’ response said that

“Section 4.2 gives the time of the data used for prediction (the year of 2020) and explains why the hours in Figure 7 are used”, but this is not so. The paper still does not specify what hours of the day were considered by the morning model and by the evening model, respectively. 

2.

Lines 153 and 316: In contrast to what the response to my original review says, the first letters of “section 5” and “equation 4” are still not capitalized.

3.

The newly added text includes several minor mistakes that should have been easily caught by simple automatic spelling and grammar checks of the manuscript:

- Line 566: The letter “s” is missing from the word “respectively”. 

- Lines 137, 142 and 309: There should be a space between the parentheses (or brackets) and the words right in front of them. 

- Line 145: There should not be a space between the word “DNN” and the subsequent comma. 

- Line 439: In referencing Figure 4, there should not be a dot between “Figure” and “4”. 

Next, some additional issues withheld from my original review are listed below. The number of issues listed below doesn’t reach 75 because quite a few issues were addressed during the revisions in response to my original review or, I guess, in response to comments by other reviewers. 

Issues about content:

Lines 33-34: It should also be mentioned that a large portion of aerosols (called “secondary aerosols”) form from gases; for example, many sulfate aerosols come from sulfur-dioxide gas emitted by cars and industrial activities. 

Lines 68-78: It should be mentioned that other geostationary satellite instruments (such as GOES ABI or MSG Seviri) also provide provide aerosol data at a high temporal resolution, but for different regions of Earth. In addition, it should also be mentioned that the EPIC imager on the DSCOVR spacecraft also provides aerosol data for the entire sunlit side of Earth (including China) at a fairly high (1-2 hour) temporal resolution. 

Lines 180-183: My understanding (based on https://www.eorc.jaxa.jp/ptree/documents/Himawari_Monitor_Aerosol_Product_v6.pdf) is that the Himawari-8 Level 3 aerosol product files do not contain cloud parameters such as cloud optical thickness, cloud top temperature, or cloud top height. The source of cloud data should be clarified.

Issues requiring clarifications:

Line 21: The expression “load band data” is unclear; either a different term should be used, or the meaning of this expression should be explained.

Line 128: It is not clear what is meant by “main activities of the system”. This should be explained, or the wording should be changed.

Line 171: The meaning of “area encryption” is not clear. This should be explained, or the wording should be changed.

Line 207: It should be clarified what PHOTONS refers to (either by text or by a URL to their website). Also, should “objects” be replaced by “sites”?

Line 214: It would help to explain the word “websites” in this context. Does Aeronet have different websites with different capabilities? Including URLs of some such websites would help. 

Lines 410-411: What is “incident gradient”? Also, a reference should be provided about the “Adam optimization algorithm”. 

Figure 4: It should be explained what the acronyms “DP”, “FC” and “ReLu” stand for, and perhaps references or descriptions should be added. As is, the figure is not understandable. Also, Lines 442-443 should explain why each layer has three sets of neurons. 

Line 450: What “other part” (part of what)?

Line 480: It should be clarified what change (in what parameter or statistics) is stable and what fluctuation (of what parameter or statistics) is small.

Line 513: It should be clarified what is meant by “about station”. Does the figure show data for a specific Aeronet station or for all Aeronet stations in Figure 1? Also, does Figure 6 show data for the morning, evening, both, or perhaps for all times of the day?

Issues about presentation:

Line 13: I recommend deleting the words “the characteristics of”.

Line 14: I recommend changing “ranges” to “makes observations at wavelengths ranging”.

Line 16 and throughout the manuscript: Please make sure there is a space after the periods marking the end of sentences.

Line 34: The word “beings” should be replaced by “activities” to clarify that it is not the people themselves who discharge the aerosols (from their own bodies), but their various activities (for example in transportation or industry). 

Line 64: The words “the sun synchronous satellite” should be replaced by “sun synchronous satellites”.

Line 65: “one” should be replaced by “once a”. 

Line 74: The letter “h” in “himawari-8” should be capitalized.

Line 96: The word “astronomical” should be deleted or replaced by something else.

Line 103: The first letter of the name of the locations is South Korea should be capitalized.

Line 127: The letter “s” should be deleted from “performs”.

Lines 218-220: The words “completed the synchronous orbit of the sun” should be replaced by “sun-synchronous orbits”. Also, “one” should be replaced by “two” (one for Terra and one for Aqua). 

Line 237: The current wording suggests that the study used the Aeronet cloud product; to avoid confusion, it should be specified that the cloud product is in fact from Himawari-8 (if this is correct).

Line 262 and everywhere else throughout the manuscript: There should be a space between the specified numbers and the units. For example, “340nm” should be replaced by “340 nm”.

Line 268: The word “points” should be replaced by “minutes”.

Lines 273-274: It should be specified that the unit of “space window radius” is pixel. 

Lines 316-317: Both occurrence of the word “meter” should be changed to “value”.

Line 327: AHI is not a satellite.

Lines 331-332: It is “solar zenith angle”, not “zenith solar angle”. It should also be clarified what “target height” and “spectral sensor” mean here. 

Lines 343 and 440: The word “soil” should be replaced by “surface”.

Lines 437-438: The list should not include the variable names in a code (such as solarAzimuth2k). Instead, it should include the physical parameters (such as “solar azimuth”). 

Line 458: The word “space” should be replaced by “area”. 

Line 460: The word “room” should be replaced, I guess by “location”.  

Figures 5, 6a, 10, and 12: The vertical axis title should be “Predicted AOD”. Vertical axis titles should also be added into Figure 7.

Line 481: It should be explained what is referred to as “common models” and which is the specific model used in Figure 5.

Line 520: The acronym is UTC, not UT. 

Line 559: The word “festival” doesn’t fit here; I it should be guess “season”.

Lines 622-625: Instead of this lengthy description, the case numbers should just be included into the panels themselves. As is, even this long description did not make it absolutely clear whether Case 2 is in the second row of the left column or in the first row of the right column. 

Line 651: The word “sampling” is redundant and should be removed.
